# Use of Technology-Based Interventions in the Treatment of Patients with Overweight and Obesity: A Systematic Review

**DOI:** 10.3390/nu12123634

**Published:** 2020-11-26

**Authors:** Lorena Rumbo-Rodríguez, Miriam Sánchez-SanSegundo, Nicolás Ruiz-Robledillo, Natalia Albaladejo-Blázquez, Rosario Ferrer-Cascales, Ana Zaragoza-Martí

**Affiliations:** 1Department of Nursing, University of Alicante, 03690 Alicante, Spain; lrr51@gcloud.ua.es; 2Department of Health Psychology, University of Alicante, 03690 Alicante, Spain; miriam.sanchez@ua.es (M.S.-S.); nicolas.ruiz@ua.es (N.R.-R.); natalia.albaladejo@ua.es (N.A.-B.); rosario.ferrer@ua.es (R.F.-C.)

**Keywords:** obesity, technologies, weight-loss, interventions

## Abstract

*Introduction*: Obesity is one of the most important health problems worldwide. The prevalence of obesity has increased dramatically in the last decades and is now recognized as a global epidemic. Given the dramatic consequences of obesity, new intervention approaches based on the potential of technologies have been developed. *Methods*: We conducted a systematic review of studies using PubMed, ScienceDirect, Cochrane Library, and MedLine databases to assess how different types of technologies may play an important role on weight loss in obese patients. *Results*: Forty-seven studies using different types of technologies including smartphones, app, websites, virtual reality and personal digital assistant were included in the review. About half of interventions (47%) found a significant effect of the technology-based interventions for weight lost in obese patients. The provision of feedback could also be effective as a complement to interventions carried out using technology to promote weight loss. *Conclusions*: The use of technologies can be effective to increase weight loss in patients with obesity improving treatment adherence through self-monitoring.

## 1. Introduction

Obesity has been defined by the World Health Organization as an abnormal or excessive accumulation of fat that poses a health risk, considered as the “epidemic of the 21st century” [1]. Obesity is one of the most important health problems, both in developed and developing countries, due to its prevalence, costs, and health effects [2]. Currently, it is accepted as a chronic and progressive disease with a high morbidity and mortality due to the likelihood of suffering comorbidities, social problems, and poor quality of life [3].

The prevalence of obesity has almost doubled worldwide over the past three decades [4]. Between 1975 and 2014, the prevalence of obesity increased from 3.2% to 10.8% in adult men, and from 6.4% to 14.9% in adult women [5].

One of the countries with the highest prevalence of obesity is the United States. This prevalence there among adults over 20 years of age is approximately 36%, with a breakdown of 38.3% for women, and 34.3% for men [4,6]. As for Spain, the prevalence of obesity among people aged 25 to 64, according to the 2014 and 2015 data from the Spanish Population Nutrition Study (ENPE), is 21.6%, and higher in men than in women, 22.8% and 20.5%, respectively [7]. Moreover, in Spain, there were around 24 million cases of overweight or obesity in 2016, that is, 70% of the adult population and three million more people than in the previous decade. If this progression continues, the figure is expected to exceed 27 million people by 2030, affecting 80% of men and 55% of women. These data are consistent with what is occurring in Europe, with moderate growth in the prevalence of excess weight in the adult population [8].

The high prevalence of obesity and its many consequences make it necessary to carry out weight-loss programs. Traditional weight-management programs with frequent face-to-face visits with the dietitian-nutritionist often take a long time, require a lot of work, and have a high cost [9]. These programs are based on monitoring diet, physical activity, and weight, reducing dietary intake, ingesting fewer kilocalories, and ultimately increasing energy expenditure by increasing physical activity [10]. These programs can be effective in the short term but have proven ineffective in the long-term, as few obese people perform these behaviors consistently, usually regaining most of the weight lost in the first year after treatment [11,12,13]. Over the past decade, it has been shown that traditional weight-management programs could be improved through the use of digital technologies or mobile applications to improve treatment adherence, recognizing that lack of sustained motivation, self-efficacy (belief in one’s ability to undertake the intervention), and lack of adherence to behavioral regimes are barriers to successful weight loss [14].

The benefits of using virtual reality, mobile phones, personal digital assistants (PDA), internet-based tools, social media, smartphones and their apps, and tablets are, on the one hand, the most direct support to achieve active patient engagement and learning. This approach, based on the potential of nutritional education and advances in food behavior theories, has been postulated to be beneficial for better health outcomes. It has also demonstrated the usefulness of such approaches to the development of self-efficacy, as suggested by preliminary evidence [15], allowing the fast recording of patient data and real-time analysis of results [16,17]. Advances in digital technology, in turn, allow for the automation of monitoring and feedback, making them more efficient and requiring less effort, both on the part of the patient who wants to lose weight and the professionals in charge of the programs for weight loss [18] Evidence to date suggests that electronic monitoring methods show higher rates of adherence to self-monitoring than traditional paper-and-pencil methods, suggesting that they can be used to facilitate greater adherence to intervention programs. At the same time, these technologies can be adapted, so it is possible to implement educational interventions and other changes in health behavior that can facilitate greater adherence to the process of self-monitoring, improve motivation for behavior change and, therefore, lead to greater success in long-term weight loss and maintenance [16,19].

Therefore, the objective of this study was to assess how different types of technologies, mobile phones, PDA, internet-based tools, social networks, smartphones and their applications, and virtual reality may play an important role on weight loss in overweight and obese patients.

## 2. Methodology and Quality Assessment

This study uses a systematic review methodology, based on the PRISMA statement. The quality of each primary study was assessed with the Cochrane Collaboration Risk of Bias (ROB) tool [20], which includes seven items covering six domains of bias. Each item is judged as having a high, low, or unclear ROB. A summary assessment is calculated based on the number of items assessed as high, low, and unclear ROB. Besides the standard summary assessment of ROB for each primary study, a summary assessment of ROB was calculated without the items related to blinding of assessor.

The first and second authors from paper rated each included article independently, and discrepancies were resolved by agreement with the third author. The Cohen’s Kappa statistic was calculated to assess interrater reliability for the ROB without items assessing blinding of participants or assessors, as all studies were rated as high ROB by the two raters when all the items were analysed.

The Cohen’s Kappa statistic was calculated to assess interrater reliability for the ROB without items assessing blinding of participants or assessors. The results showed an agreement between two raters between 0.6 to 0.85.

### 2.1. Data Sources

The systematic search was carried out in the PubMed, ScienceDirect, Cochrane Library, and MedLine databases. Additional articles were identified by searching the references of another article.

### 2.2. Search Strategy

The search strategy aimed to identify the published studies available in full text. A bulk search strategy was used, using both the MeSH descriptors and terms in the titles or abstracts, which were as follows: “virtual reality exposure therapy”, “virtual reality”, “virtual world”, “virtual environment”, “3D vision”, “smartphone”, “cell phone”, “technology”, “obesity”, “overweight”, “diet therapy”, and “weight loss” joined by Boolean operators (AND, OR) as follows: (virtual reality exposure therapy OR virtual reality OR virtual world OR virtual environment OR 3D vision OR smartphone OR cell phone OR technology) AND (obesity OR overweight) AND (diet therapy OR weight loss). The date of the last search was 26 February 2020, and no time restrictions were made about the year of publication of the studies. Table 1 shows the search strategy used in the Pubmed database.

### 2.3. Selection of Articles

Abstracts identified through the bibliographic search were independently evaluated by two authors to confirm the inclusion criteria. The quality of each study was independently evaluated by two authors, using the Crombie criteria adapted by Petticrew and Roberts [21]. Disagreements were resolved by a third author.

### 2.4. Inclusion and Exclusion Criteria

Inclusion criteria were: (I) articles that were available in full text and written in English or Spanish; (II) articles whose participants were 18 years of age or older, overweight or obese; (III) articles that presented at least two groups for comparison, one of them with a weight management intervention through some type of technology; and (IV) articles in which the weight was reported with a numerical value before, during, and/or after the intervention.

The exclusion criteria were: (I) articles not related to the subject of the study or articles that were protocols of intervention without results; (II) articles that were reviews or meta-analyses; (III) documents that were summaries for conferences; and (IV) presence of major pathologies such as mental illnesses, eating disorders, and cancer.

### 2.5. Extracted Data

Data extraction was carried out by the lead author of the review, taking into account the year of publication (2009–2020), design and objective of the study, sample size, number of groups in the study, participants’ mean age, country of origin, type and duration of the intervention, type of technology used in weight loss and/or maintenance programs, weight changes and other relevant intervention results, such as adherence to dietary-nutritional treatment.

### 2.6. Synthesis of Results

After completing data extraction, the results were grouped according to the technology used in four large main blocks (1. Smartphone, 2. Virtual Reality, 3. Website, and 4. PDA or Electronic Journal (EJ), and the results were compared regarding weight changes (pre- and post-intervention) and adherence. In turn, there was a fifth group of studies that did not use any of the aforementioned types of technology, which has the following name: 5. Other types of technology. Finally, a table was designed that showed the average weight lost by each type of technology.

## 3. Results

In total, 2403 studies were identified. After the duplicates were removed (*n* = 1063), the titles and summaries were read, and another 1293 articles were deleted according to the different exclusion criteria. Finally, 47 articles were included in this review (Figure 1).

### 3.1. Descriptive Data and Types of Studies

Table 2 shows the characteristics of the articles included. Of the participants, 51.08% were women and the remaining 48.77% were men, with a mean participant age of about 40.9 years. 

As for the country of origin, almost 72.5% of the articles were performed in the United States (*n* = 34) [11,15,16,17,23,24,25,26,27,29,30,31,34,35,37,38,39,40,41,42,45,46,47,49,52,53,55,56,57,58,59,60,61,62], three studies were conducted in Australia [36,48,51], and two in Spain [13,33]. Three and two articles, respectively, were carried out in China and the United Kingdom [22,32,43,44,54]. And finally, one study was carried out in Italy, one in Korea, and one in Iran [12,28,50].

Table 2, which also lists the design of the studies, shows that of the 47 studies included, 30 of them were randomized controlled trials [11,12,13,23,25,27,29,30,31,32,33,35,36,38,39,40,41,44,45,47,48,50,51,52,53,54,55,57,61,62]. Six were randomized pilot studies [17,26,32,37,49,56] and one was a non-randomized pilot study (16). Six were randomized pretest-posttest design [24,34,38,46,58,60] and four were quasi experimental designs [15,22,28,42].

### 3.2. Type of Technology and Provision of Feedback

#### 3.2.1. Smartphone

Table 3 shows the type of technology that was used in each of the studies included. Among all the studies, 28 of them used a smartphone in the intervention for weight loss and/or maintenance. In eighteen studies a mobile App (App) was used in their intervention; eleven of them used an App to monitor intake, weight, and/or physical activity [11,13,16,22,24,25,26,27,28,29,30]. Five articles, in addition to an App to monitor intake and weight, used a physical activity monitor [17,23,31,33,34]. One article used the mobile app and/or the website [32], and in another article, two Apps were used, one for self-monitoring and one for video-conferences [35]. In the other 10 articles, the intervention was carried out by telephone calls and/or SMS [36,37,38,39,40,41,42,44,45].

As for whether or not feedback was available (i.e., whether or not participants received some kind of response, either via email, short text message or SMS, or phone call), only one study did not present this aspect [22].

#### 3.2.2. Virtual Reality

Four studies (Table 3) included an intervention based on the potential of virtual reality techniques, using 3D avatars (Second Life or similar ones) [12,15,46,47]. Two of the four articles did not provide feedback [15,47].

#### 3.2.3. Website

In 4 articles, the intervention was carried out with the help of a website; two of them used it to monitor dietary intake, weight, and/or physical activity [50,51]. One article used it for video-conferences [31] and another study, in addition to using a website, used a pedometer [48] (Table 3). All of them provided feedback.

#### 3.2.4. Personal Digital Assistant (PDA) or Electronic Journal (EJ)

Four articles used a (PDA) or an electronic journal (EJ) to monitor dietary intake, weight, and/or physical activity, and all of them provided feedback [52,53,54,55].

#### 3.2.5. Other Types of Technology

Seven of the articles analyzed used a different type of technology. Of these seven, three articles used a physical activity monitor [56,57,58], two of them used online software [59,60], one study used DVDs [61], and the last one used Facebook, an App, text messages, emails, a website, and technology-mediated communication with a health coach [62]. Of these articles, only one of them did not provide feedback [41].

### 3.3. Effectiveness of Each Type of Technology in Weight Loss and/or Maintenance and Adherence

#### 3.3.1. Smartphone

APPs

Six of the articles that used or supplemented the intervention with an App exclusively or together with some physical activity monitor, reported evidence of weight loss compared to the control or comparison groups (Table 4) [28] (*p <* 0.05); [11] (*p <* 0.05); [31](*p <* 0.001); [17] (*p =* 0.042); [26] (*p =* 0.026); [41] (*p <* 0.05). In He et al.’s work. [22], the differences were significant for men (*p <* 0.001) but not for women. In contrast, in five articles, the control or comparison groups lost more weight, but the differences did not reach significance in any of them [16] (*p =* 0.19); [47] (*p =* n.s); [34] (*p =* 0.0997); [23] (*p =* n.s); [24] (*p* > 0.05) Finally, in one study, both groups lost about 2.6 kg (*p =* 0.88) [27]. The average weight lost by the App groups was 3.82 kg, with 7.9 kg and 0.03 kg, respectively, the highest and lowest average amounts lost.

In seven articles, adherence was greater in the groups that used or complemented the intervention with an App, with significant differences in three of them [32] (*p <* 0.001); [17] (*p <* 0.05); [23] (*p <* 0.001). In Spring et al.’s work [23], adherence to self-monitoring contributed to weight loss (*r* (84) = 0.36–0.51, *p <* 0.001) and in that of Ross and Wing. [17], the percentage change in weight was significantly associated with adherence to intake control (*r* = −0.48, *p <* 0.001) and weight (*r* = −0.42, *p =* 0.085). Only in the work of Rogers et al. [34], the standard group monitored their diet for a longer average number of days than the technology groups.

SMS and/or calls

Seven of the studies analyzed that used SMS or calls found a significant effect of their use, with significant weight loss compared to the control or comparison groups [44] (*p =* 0.006); [45] (*p =* 0.02); [42] (*p =* n.s); [37] (*p =* 0.09); [43] (*p <* 0.0001); [35] (*p =* 0.03); [36] (*p =* 0.01). Only in one study did the comparison group lose more weight, [39] (*p =* 0.002). The SMS or call groups lost an average of 3.07 kg, with 4.87 kg and 1.27 kg being the highest and lowest average amounts lost.

As for adherence, two of the studies reported the existence of an association between increased adherence and greater weight loss [37,38]. In contrast, in another study, weight loss at 6 months did not correlate with the total of the follow-up days (*r* = 0.14, *p =* 0.27) [43]. In four studies, it was noted that despite the inclusion of technology, adherence decreased over time [40,42,44,45]. For example, in one study, it was 66% at the start of the intervention and 52% at the end of the intervention [42].

#### 3.3.2. Virtual Reality

Table 4 shows two studies where groups that used virtual reality lost more weight, with significant group differences in one of them [47] (*p =* 0.04). On the other hand, in two articles [12,46], the control groups lost more weight, and the differences were significant in one of them [46] (*p <* 0.05). However, in this same study [46], weight maintenance was significantly higher in the virtual reality group (14% vs. 9.5%, *p <* 0.05). Manzoni et al. [12] stated that the virtual reality group was more likely to further maintain or improve weight loss at a one-year follow-up. Groups that used virtual reality lost an average of 4.7 kg, with 7.3 kg and 0.79 kg being the highest and lowest average amounts lost.

#### 3.3.3. Website

Table 4 shows that all groups that used a website lost significantly more weight than the control or comparison groups [48] (*p <* 0.001); [50] (*p =* 0.001); [49] (*p =* 0.0002), except for one group where the differences did not reach significance [51] (*p =* 0.408). The average weight loss was 3.75 kg, with 5.3 kg and 1.4 kg being the highest and lowest weight amounts lost by the groups that used a website.

In terms of adherence, one study observed significant correlations between weight change at 12 months and the number of days of diet entries (*r* = 0.69; *p <* 0.001), number of daily exercise entries (*r* = 0.54; *p* = 0.004), and number of weekly weight entries (*r* = 0.56; *p =* 0.004) [51]. In another study, each additional target set and each weight measurement recorded were associated with greater weight loss, of 0.32 kg and 0.21 kg, respectively [48].

#### 3.3.4. Personal Digital Assistant (PDA) or Electronic Journal (EJ)

Among the four studies that used this type of technology, just in one of them, the PDA group lost significantly more weight than the control group (−2.9 kg vs. −0.02 kg; *p =* n.s.) [55]. The groups that used this technology lost an average of 2.0 kg, with 2.9 kg and 1.18 kg being the highest and lowest average amounts lost by the PDA groups.

In two studies, adherence was higher in the PDA groups [52]. However, it decreased as of the third week [52]. Wang et al. [53] stated that, compared to the paper control, using PDA to control diet (*p =* 0.027) and physical activity (*p =* 0.014) had significant direct effects on weight loss.

#### 3.3.5. Other Types of Technology

As noted in Table 4, in all the studies, the technology groups lost more weight than the control or comparison groups [56,57,58,59,60,61,62], with significant differences in four of them [60] (*p <* 0.05); [57] (*p <* 0.0001); [58] (*p =* 0.044); [59] (*p =* 0.02). Physical activity monitors were the type of technology that achieved the highest weight loss (6.21kg), while DVDs were the one that achieved the lowest weight loss (0.48 kg) (Appendix A). Adherence was higher in the technology groups [56,58,60] and, in two works, it was significantly related to weight loss [60] (*r* = 0.24, *p* ≤ 0.05); [58] (*r* = 0.57, *p <* 0.001).

## 4. Discussion

The results found in this work indicate that weight loss was greater in the groups whose intervention was performed or complemented by one of the aforementioned types of technology, although in 13 studies [13,15,26,29,32,33,38,51,52,54,56,61,62], the differences with the control or comparison groups were not statistically significant. However, the same cannot be concluded regarding weight maintenance, since most of the studies did not include this outcome. In another study, the two groups that used a smartphone lost more weight than the control group, with significant differences between the SMS + Coaching group and the control group but not between the SMS group alone and the control group [40]. At the same time, adherence was better in the technology groups, except for one study where the standard group monitored their diet for a greater average number of days than the technology groups [34]. Except for two studies [11,43], adherence was associated with the weight changes that took place in the technology groups, and in all the works in which the association was studied, it was observed that greater adherence led to greater weight loss.

In general, these results suggest that the use of different types of technology for self-monitoring of diet, physical activity, and/or weight is effective in promoting weight loss among people who are overweight or obese.

In this systematic review, about half of interventions (47%) performed or complemented by some kind of technology helped participants to achieve significant weight losses, compared to the control or comparison groups. These results follow a very similar line to those obtained in other systematic reviews. For example, Raaijmakers et al. [63] found that half of the technology-based interventions (54%) significantly helped participants lose weight, compared to the lack of attention or habitual attention. Similarly, Allen et al. [64] found that in more than half of the studies analyzed (53%), statistically significant weight loss was evident in the intervention group, compared to that of the control group. Another study comparing weight changes between eHealth interventions and control groups without intervention found a significantly greater decrease in weight in eHealth interventions (*M* = −2.70 kg, *p* < 0.00001) than in the control group. Also, when comparing eHealth interventions and control groups that received a minimal intervention, a significantly greater decrease was observed in eHealth interventions (*M* = −1.40 kg, *p* < 0.0001) [65].

The provision of feedback could be effective as a complement to interventions carried out using technology to promote weight loss [52,53]. This might suggest that receiving feedback in the form of text messages or emails could improve adherence to self-monitoring and, as a result, lead to increased weight loss. Nevertheless, more research is needed on this topic since the evidence found in this systematic review is not strong enough. In one study, adherence to self-monitoring was higher in those receiving feedback (78%) compared to those who did not receive it (78% vs. 64%; *p <* 0.001), and also, participants who received feedback lost more weight than those who did not receive it (7.0 kg vs. 5.0 kg (*p* < 0.05) [66]. It has also been observed that participants who received personalized feedback had an average weight loss of 2.13 kg more (*p* < 0.00001) at 3 and 6 months, compared to the control groups. However, this was not observed in interventions lasting 12 months or more [67].

Based on the results obtained, using some kind of technology also implies that people who are overweight or obese will show better adherence to treatment because the new technologies allow a much faster and more efficient recording of data related to dietary intake, physical activity, and weight, as well as their analysis in real time [17,18,19,68]. Semper et al. [69] stated that participants who used an App were more likely to adhere to the self-monitoring of dietary intake.

Of all the methods analyzed, physical activity monitors were the type of technology that achieved the greatest weight loss (*M* = 6.21 kg), followed by virtual reality (4.7 kg), website (3.75 kg), smartphone (3.44 kg), and PDA (2.0 kg) (Appendix A). However, only seven studies of those analyzed used these types of technology to perform the intervention, and therefore, it is impossible to know exactly whether these mean weight losses would remain so high after being evaluated in more groups of people.

This review presents a series of limitations. First, the wide variability in the design of the studies included limits the conclusions that can be reached. Second, the search only included English and Spanish publications, which may not have represented all the available evidence. Thirdly, heterogeneity of the time periods of the intervention was high, ranging from a few weeks to 24 months, which can affect the strength of our results and conclusions. Fourth, the presence of studies that used a small sample size may be associated with greater uncertainty about the measured effect. And, fifth, heterogeneity in the type of intervention performed and the groups with which the comparison was made, which can make the comparison of effectiveness difficult to investigate. However, it has a main strength, which is the fact that it is one of the few systematic reviews that encompasses studies that used different types of technology to carry out the intervention and it does not focus solely on one of them.

## 5. Conclusions

Weight loss programs for people who are overweight or obese, carried out or supplemented by some kind of technology, seem to lead to greater weight loss compared to traditional programs. Physical activity monitors and virtual reality were the types of technology that lead to increased weight loss, although further research is needed on the use of these types of technology, as the evidence found is scarce. The use of technology also seems to allow improvement in adherence to treatment, as it allows a simpler and faster self-monitoring. In addition, although more research is needed, this could improve more when the technology is accompanied by immediate feedback. However, future research should focus on this, as, despite the use of technology, adherence to dietary-nutritional treatment often decreases over time, resulting in less weight loss as time passes.

Finally, research on this issue should continue to be carried out, as overweight and obesity are currently very present worldwide, and also, as technologies are part of the day-to-day life of today’s society, these could be of great help in weight loss programs, as suggested by the results of this systematic review.

## Figures and Tables

**Figure 1 nutrients-12-03634-f001:**
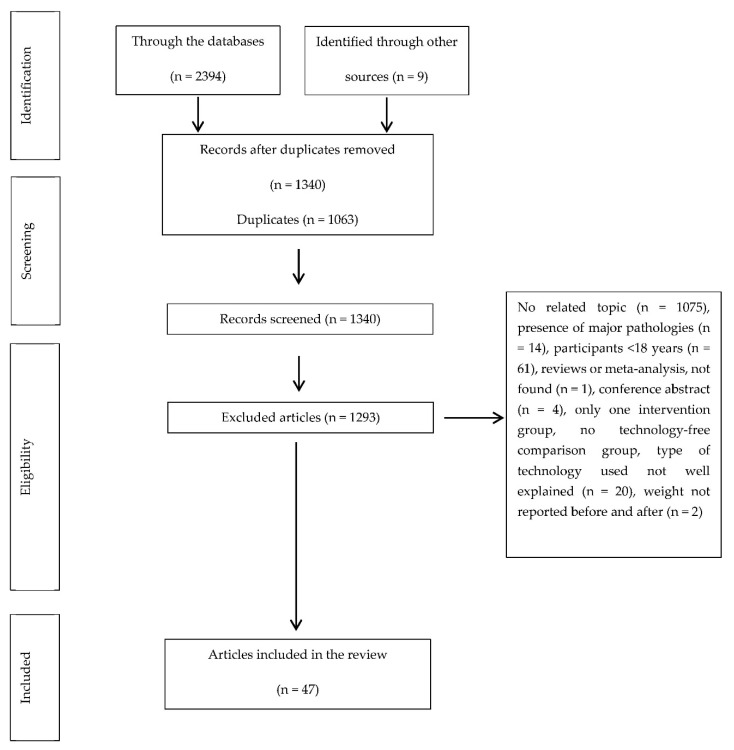
Selection of the studies.

**Table 1 nutrients-12-03634-t001:** Pubmed Search Strategy.

Search Strategy	
**#1.**	(“virtual reality exposure therapy” [MeSH * Terms] OR “virtual reality” [Title/Abstract] OR “virtual world” [Title/Abstract] OR “virtual environment” [Title/Abstract] OR “3D vision” [Title/Abstract] OR “smartphone” [Title/Abstract] OR “cell phone” [Title/Abstract] OR “technology” [Title/Abstract])
**#2.**	(“obesity” [MeSH Terms] OR “obesity” [Title/Abstract] OR “overweight” [MeSH Terms] OR “overweight” [Title/Abstract])
**#3.**	(“diet therapy” [MeSH Terms] OR “diet therapy” [Title/Abstract] OR “weight loss” [MeSH Terms] OR “weight loss” [Title/Abstract])
**#4.**	#1. AND #2. AND #3.

*: Medical Subject Headings.

**Table 2 nutrients-12-03634-t002:** Description of the studies included in the review.

Authors, Year, [Reference]	Country	Year	Mean Age (Years)	Nr Sample (n)	Objective	Study Type
He et al., 2017, [22]	China	2017	37	Total: 15,310	Assess the effectiveness of a mobile app (WeChat) as an intervention in weight loss behavior.	Cohort study
				Men: 9282		
				Women: 6028		
Spring et al., 2017 [23]	USA	2017	39.3	Total: 96	Determine the effects on weight loss from three brief behavioral weight loss interventions with and without coaching and mobile technology.	Three-month randomized controlled efficacy study of weight loss
				Men: 15		
				Women: 81		
Ross & Wing., 2016 [17]	USA	2016	51.1	Total: 80	Investigate the impact of new self-monitoring technology (compared to traditional self-monitoring tools), provided with and without brief telephone intervention, on weight loss in overweight and obese adults	Randomized pilot study
				Men: 11		
				Women: 69		
Apiñaniz et al., 2019 [13]	Spain	2019	38.55	Total: 110	Evaluate the effectiveness of a mobile app to support the provision of health advice for weight loss	Randomized controlled trial
				Men: 31		
				Women: 79		
Wharton et al., 2014 [16]	USA	2014	42	Total: 47	Evaluate the use of a smartphone app for diet and weight loss self-monitoring against traditional diet counseling and self-monitoring methods.	Pilot study
				Men: 12		
				Women: 35		
Thomas et al., 2019 [24]	USA	2019	55.1	Total: 276	Determine whether the weight losses of a smartphone-based (SMART) behavioral obesity treatment differed from those of more intensive group behavioral obesity treatment (GROUP) and a control condition (CONTROL).	Randomized clinical trial
				Men: 47		
				Women: 229		
Svetkey et al., 2015 [25]	USA	2015	29.4	Total: 365	Determine the effect on weight of two mobile technology-based (mHealth) behavioral weight loss interventions in young adults.	Randomized controlled trial
				Men: 111		
				Women: 254		
Allen et al., 2013 [26]	USA	2013	44.9	Total: 68	Assess the feasibility, acceptability, and preliminary effectiveness of theory-based behavioral interventions performed with smartphone technology.	Randomized controlled pilot study
				Men: 15		
				Women: 53		
Turner-McGrievy & Tate., 2011 [27]	USA	2011	42.9	Total: 96	Examine whether a combination of podcasting, mobile support communication, and mobile diet monitoring can help people lose weight.	Randomized trial
				Men: 24		
				Women: 72		
Lee et a., 2010 [28]	Corea	2010	28.85	Total: 36	Evaluate the effectiveness of the SmartDiet mobile phone app compared with the acquisition of dietary information, weight management, and user satisfaction.	Case study control
				Men: -		
				Women: -		
Laing et al., 2014 [29]	USA	2014	43.3	Total: 212	Assess the effect of providing this application of a free and widely used smartphone for weight loss to patients in their primary care clinic.	Randomized controlled trial
				Men: 154		
				Women: 58		
Napolitano et al., 2013 [11]	USA	2013	2 0.47	Total: 52	Evaluate the feasibility, acceptability and preliminary effectiveness of a novel technology-based weight loss intervention for college students using adapted evidence-based weight loss content.	Randomized controlled trial
				Men: 7		
				Women: 45		
Stephens et al., 2017 [30]	USA	2017	20	Total: 62	Evaluate the effectiveness of a smartphone app, based on behavior for weight loss combined with text messages from a weight-health coach, body mass index, and waist circumference in young adults compared to a control condition.	Randomized controlled trial
				Men: 18		
				Women: 44		
Martin et al., 2015 [31]	USA	2015	44.4	Total: 40	Evaluate the effectiveness of SmartLoss, a smartphone-based weight-loss intervention.	Randomized controlled trial
				Men: 7		
				Women: 33		
Carter et al., 2013 [32]	Reino Unido	2013	42	Total: 128	Report the acceptability and feasibility of a weight management self-monitoring intervention performed with a smartphone app, compared to a website and a written diary.	Randomized controlled pilot study
				Men: 29		
				Women: 99		
Hernández-Reyes et al., 2020 [33]	Spain	2020	41.5	60 Women	Evaluate the effectiveness of "push notifications" in an intervention to improve body composition of overweight or obese adult women, through a dietary procedure, and analyze the evolution of body composition based on push notifications and prescribed physical activity (PA).	Randomized controlled trial
Rogers et al., 2016 [34]	USA	2016	39.9	Total: 39	Compare a group of behavioral weight loss in-person intervention with technological interventions in obese adults.	Randomized pretest posttest design
				Men: 8		
				Women: 31		
Alencar et al., 2019 [35]	USA	2019	46.8	Total: 25	Determine the effectiveness of a medically controlled weight management program with weekly health training versus no health training through weight loss video-conferences using mobile health devices.	Randomized controlled trial
				Men: 12		
				Women: 13		
Lewis et al., 2019 [36]	Australia	2019	49	Total: 61	Investigate the effectiveness and optimal timing of using phone calls and text messages as complementary tools to support a community obesity management program.	Randomized cross-controlled trial
				Men: 14		
				Women: 47		
Steinberg et al., 2013 [37]	USA	2013	38.3	50 Women	Assess the feasibility of a text message intervention for weight loss among predominantly Black women.	Randomized controlled pilot study
Shapiro et al., 2012 [38]	USA	2012	41.9	Total: 170	Evaluate a daily intervention of weight loss through text messages	Randomized controlled trial
				Men: 111		
				Women: 59		
Jakicic et al., 2016 [39]	USA	2016	3 0.9	Total: 470	Assess whether adding portable technology to a behavioral intervention would improve weight loss for 24 months among 18- to 35-year-olds	Randomized controlled trial
				Men: 136		
				Women: 334		
Godino et al., 2019 [40]	USA	2019	41.7	Total: 298	Examine the effectiveness of a 1-year intervention designed to reduce weight among overweight and obese English- and Spanish-speaking adults via SMS (ConTxt) alone or in combination with short monthly health coaching calls.	Randomized controlled trial
				Men: 70		
				Women: 228		
Newton et al., 2018 [41]	USA	2018	56	Total: 97	Evaluate the feasibility (tolerance and satisfaction of SMS text messages) and the effectiveness (compared to a control group) of using SMS text messages to promote weight loss in African-American adults enrolled in a church-based weight loss program.	Randomized controlled trial
				Men: 8		
				Women: 89		
Bouhaidar et al., 2013 [42]	USA	2013	45	Total: 28	Evaluate the effect of personalized text messages on body weight change in overweight and obese adults in a community-based weight management program.	Quasi-experimental analysis with pretest and posttest
				Men: 2		
				Women: 26		
Lin et al., 2014 [43]	China	2014	38.21	Total: 123	Test the impact on weight change of a trial of a 6-month text-assisted weight loss intervention in overweight adults in Beijing.	Randomized clinical trial
				Men: 49		
				Women: 74		
Haapala et al., 2009 [44]	United Kingdom	2009	38	Total: 124	Investigate short- and long-term effectiveness and weight loss predictors in a mobile phone weight loss program among healthy overweight adults.	Randomized controlled trial
				Men: 28		
				Women: 96		
Patrick et al., 2009 [45]	USA	2009	44.9	Total: 65	Determine whether text messages were a useful and effective strategy to help adults control their weight and improve their results.	Randomized controlled trial
				Men: 13		
				Women: 52		
Sullivan et al., 2013 [46]	USA	2013	31.1	Total: 20	Evaluate Second Life’s effectiveness for weight loss and maintenance	Randomized trial
				Men: 3		
				Women: 17		
Behm-Morawitz et al., 2016 [47]	USA	2016	25	Total: 92	Examine the effectiveness of virtual realization and play in a virtual social world (Second Life) to increase health self-efficacy among overweight adults	Randomized controlled trial
				Men: 2		
				Women: 90		
Johnston et al., 2012 [15]	USA	2012	41.9	Total: 54	Explore the effectiveness of a weight loss program in the virtual world in relation to weight loss and behavior change	Quasi experimental
				Men: 9		
				Women: 45		
Manzoni et al., 2016 [12]	Italy	2016	35.63	163 Women	Test the long-term effectiveness of enhanced cognitive behavioral obesity therapy (CBT), which includes a virtual reality (VR) module aimed at unlocking the body’s negative memory and modifying its behavioral and emotional correlates	Randomized controlled trial
Morgan et al., 2014 [48]	Australia	2014	47.3	159 Men	Provide a comprehensive assessment of the SHED-IT (self-help, exercise and diet using information technology) weight loss program process for men	Randomized trial, controlled blinded by the evaluator
Azar et al., 2015 [49]	USA	2015	46.3	64 Men	Assess whether group interventions using web-based group video conference (VC) technology are effective for weight loss.	Randomized pilot study
Abdi et al., 2015 [50]	Iran	2015	42	Total: 435	Evaluate the effect of intervention based on new communication technologies and social cognitive theory on weight management in government employees in the city of Hamadan, western Iran, in 2014.	Randomized controlled trial
				Men: 125		
				Women: 310		
Morgan et al., 2011 [51]	Australia	2011	35.9	65 Men	Examine the 12-month results of the SHED-IT Internet-based program (Self-Help, Exercise, and Diet Using Information Technology) for overweight men to determine whether they could maintain weight loss.	Randomized controlled trial
Burke et al., 2012 [52]	USA	2012	46.8	Total: 210	Determine whether diet monitoring using only one PDA or PDA+daily personalized feedback was superior to using a conventional paper diary on weight loss and weight maintenance in a 24-month study. Examine whether weight loss would be greater for those who adhere to self-control (monitoring).	Randomized controlled trial
				Men: 32		
				Women: 178		
Wang et al., 2012 [53]	USA	2012	46.8	Total: 210	Report the mediation effect of self-monitoring diet adherence and physical activity on weight loss in overweight or obese participants after 12 months in a technology-based behavioral weight loss program.	Randomized clinical trial
				Men: 32		
				Women: 178		
Chung et al., 2014 [54]	China	2014	37.4	Total: 60	Test the effectiveness of dietary TV as an intervention for weight reduction.	Double-blind randomized controlled trial
				Men: 22		
				Women: 38		
Spring et al., 2013 [55]	USA	2013	57.7	Total: 69	Test whether a connective mobile technology system, telephone training and standard treatment for obesity improved weight loss outcomes compared to standard group treatment alone for obesity.	Randomized controlled trial
				Men: 59		
				Women: 10		
Unick et al., 2012 [56]	USA	2012	42.4	Total: 29	Examine whether the addition of portable PA monitors to the SBWL (standard behavioral weight loss program) treatment for people with severe obesity improved PA and self-control after a 6-month intervention.	Randomized pilot study
				Men: 5		
				Women: 24		
Shuger et al., 2011 [57]	USA	2011	46.8	Total: 197	Determine the effectiveness of continuous self-monitoring and technology feedback (achieved through SWA automation) alone and in combination with GWL (group weight loss) to improve weight loss and reduce waist circumference over a 9-month period in sedentary overweight or obese adults.	Randomized controlled trial
				Men: 36		
				Women: 161		
Pellegrini et al., 2012 [58]	USA	2012	44.2	Total: 51	Evaluate the effectiveness of a technology-based system (TECH) in weight loss when used alone or in combination with a 6-month in-person behavioral weight loss intervention in overweight and obese adults.	Randomized trial
				Men: 7		
				Women: 44		
Dunn et al., 2014 [59]	USA	2014	48.96	Total: 1711	Compare the effectiveness of online delivery of a weight management program using synchronous (real-time) distance education technology with in-person delivery.	Pre- and post-study
				Men: 173		
				Women: 1538		
Chambliss et al., 2011 [60]	USA	2011	45	Total: 120	Develop and evaluate a 12-week weight management intervention that involves computer self-monitoring and technology-assisted feedback with and without an improved behavioral component.	Randomized trial
				Men: 21		
				Women: 99		
Ing et al., 2018 [61]	USA	2018	45.95	Total: 217	Compare the effectiveness of a 9-month workplace-based weight loss maintenance intervention delivered on DVD versus face-to-face.	Randomized controlled trial
				Men: 28		
				Women: 189		
Godino et al., 2016 [62]	USA	2016	22.7	Total: 404	Evaluate the effectiveness of a 2-year, theory-based weight loss intervention that was delivered remotely and adaptively through user experiences integrated with Facebook, mobile apps, text messages, emails, a website, and technology-mediated communication with a health coach.	Randomized controlled trial
				Men: 120		
				Women: 284		

**Table 3 nutrients-12-03634-t003:** Characteristics of the intervention.

Authors, Year, [Reference]	Intervention	Duration Intervention	Nr. of Groups	Technology	Feedback
He et al., 2017 [22]	Diet and physical activity	6 months	2 (technology group and control group)	Mobile App (WeChat)	Yes
Spring et al., 2017 [23]	Diet and physical activity	12 months	3 (self-guided, standard and technology)	Mobile App + Accelerometer	Yes
Ross & Wing., 2016 [17]	Diet and physical activity	6 months	3 (standard, technology and technology + phone)	App (Fibit), Fibit Activity Monitor Zip, Fibit Aria Smart Scale	Yes
Apiñaniz et al., 2019 [13]	Diet and physical activity	6 months	2 (control group and intervention group)	Mobile App (Aktidiet)	Yes
Wharton et al., 2014 [16]	Diet and physical activity	8 weeks	3 (control group (paper), “memo” group (mobile notes) and app)	Mobile App (“Lose it”)	Yes
Thomas et al., 2019 [24]	Diet and physical activity	18 months	3 (control group, “Group” sessions, and “Smart” sessions + App)	App (MyFitnessPal)	Yes
Svetkey et al., 2015 [25]	Diet and physical activity	24 months	3 (control group, cell phone group (CP) and “personal coaching” group (PC)	CP: App PC: Smartphone (self-monitoring)	Yes
Allen et al., 2013 [26]	Diet and physical activity	6 months	4 (intensive counseling group (IC), intensive counseling group + smartphone (IC + SP), less intensive counseling group + smartphone (LIC + SP) and smartphone group (SP)	App (Lose It!)	Yes
Turner-McGrievy & Tate., 2011 [27]	Diet and physical activity	6 months	2 (Podscat group and Podscat+Mobile group)	Podscat and mobile (App FatSecret’s)	Yes
Lee et a., 2010 [28]	Diet and physical activity	6 weeks	2 (control group and intervention group)	App (SmartDiet)	No
Laing et al., 2014 [29]	Diet and physical activity	6 months	2 (control group and intervention group)	App (MyFitnessPal)	Yes
Napolitano et al., 2013 [11]	Diet and physical activity	8 weeks	3 (Facebook group, Facebook plus group, and control group)	App (Facebook)	Yes
Stephens et al., 2017 [30]	Diet and physical activity	3 months	2 (Smartphone+health coach group and control group)	App (Lose It!)	Yes
Martin et al., 2015 [31]	Diet and physical activity	12 weeks	2 (technology group and “health education” group)	Mobile app, weight scale and activity measurer	Yes
Carter et al., 2013 [32]	Diet and physical activity	6 months	2 (intervention group and face-to-face group)	Mobile App (My Meal Mate) Website	Yes
Hernández-Reyes et al., 2020 [33]	Diet and physical activity	6 months	2 (control group and intervention group)	App + pedometer	Yes
Rogers et al., 2016 [34]	Diet and physical activity	6 months	3 (standard group, TECH group and EN-TECH) group	Smartphone+physical activity monitor	Yes
Alencar et al., 2019 [35]	Diet and physical activity	12 weeks	2 (control group and intervention group)	“American Well” App: Video-conference MyFitnessPal App: self-monitoring	Yes
Lewis et al., 2019 [36]	Diet, physical activity, stress/mood and sleep- and lifestyle-related habits.	8 months	2 (technology and control group)	Phone (calls and text messages)	Yes
Steinberg et al., 2013 [37]	Diet and physical activity	6 months	2 (intervention group and control group)	Automated system which includes daily text messages.	Yes
Shapiro et al., 2012 [38]	Diet and physical activity	12 months	2 (control group and intervention group)	Phone (SMS and MMS (multimedia messaging service)	Yes
Jakicic et al., 2016 [39]	Diet and physical activity	24 months	2 (SWBI group and EWLI group)	Phone sessions, messages, website, PA monitor	Yes
Godino et al., 2019 [40]	Diet and physical activity	12 months	3 (control group, SMS group, and SMS + coaching group)	SMS	Yes
Newton et al., 2018 [41]	Diet and physical activity	6 months	2 (control group and intervention group)	SMS	Yes
Bouhaidar et al., 2013 [42]	Diet and physical activity	12 weeks	2 (control group and intervention group)	SMS	Yes
Lin et al., 2014 [43]	Diet and physical activity	6 months	2 (control group and intervention group)	Text messages	Yes
Haapala et al., 2009 [44]	Diet and physical activity	12 months	2 (control group and intervention group)	SMS	Yes
Patrick et al., 2009 [45]	Diet and physical activity	4 months	2 (comparison group and intervention group)	SMS	Yes
Sullivan et al., 2013 [46]	Diet and physical activity	9 months	2 (virtual reality group and face-to-face group)	Virtual Reality (Second Life)	Yes
Behm-Morawitz et al., 2016 [47]	Diet and physical activity	4 weeks	3 (3D group, 2D group, and non-intervention group)	3D virtual world, 2D website	No
Johnston et al., 2012 [15]	Diet and physical activity	12 weeks	2 (virtual reality group and face-to-face group)	Virtual reality (Club One Island via Linden Lab’s Second Life)	No
Manzoni et al., 2016 [12]	Diet and physical activity	12 months	3 groups (SBP group, SBP-CBT group, and SBP + VR-CBT group)	Virtual reality (NeuroVR open source software)	Yes
Morgan et al., 2014 [48]	Diet and physical activity	6 months	3 (“shed it resources” group, “shed it online”, and control)	Web plus pedometer	Yes
Azar et al., 2015 [49]	-	3 months	2 (control group and intervention group)	Video conferencing technology via web	Yes
Abdi et al., 2015 [50]	Diet and physical activity	9 months	3 (control group, web group, phone group)	Website (“Healthy employee), cell phone	Yes
Morgan et al., 2011 [51]	Diet and physical activity	12 months	2 (internet group and control group)	Web	Yes
Burke et al., 2012 [52]	Diet and physical activity	24 months	3 (paper *diary* group, PDA group, and PDA + Feedback group)	PDA	Yes
Wang et al., 2012 [53]	Diet and physical activity	12 months	3 (paper *diary* group, PDA group, and PDA + Feedback group)	PDA	Yes
Chung et al., 2014 [54]	Diet	12 weeks	3 (“food diary” group, “electronic diary” group, and control group)	Electronic diary	Yes
Spring et al., 2013 [55]	Diet and physical activity	12 months	2 (standard group and +mobile group)	PDA	Yes
Unick et al., 2012 [56]	Physical activity and diet changes	6 months	2 (SBWL group and SBWL + Technology group	Sensewear bracelet	Yes
Shuger et al., 2011 [57]	Diet and physical activity	9 months	4 (standard group, GWL group, SWA group, and GWL+SWA group)	SenseWear Armband	Yes
Pellegrini et al., 2012 [58]	Diet and physical activity	6 months	3 (SBWL group, SBWL+TECH group, and TECH group)	Monitoring armband	Yes
Dunn et al., 2014 [59]	Diet and physical activity	15 weeks	2 (online group and in-person group)	Online teaching software (Elluminate Live!)	Yes
Chambliss et al., 2011 [60]	Diet and physical activity	12 weeks	3 (basic feedback group, enhanced feedback group, and control group)	Software (BalanceLog)	Yes
Ing et al., 2018 [61]	Diet and physical activity	9 months	2 (DVD group and face-to-face group)	DVDs	No
Godino et al., 2016 [62]	Diet and physical activity	24 months	2 (Smart group and control group)	Facebook, App, text messages, emails, a website and technology-mediated communication with a health coach	Yes

**Table 4 nutrients-12-03634-t004:** Results of the intervention regarding changes in weight and adherence.

Authors, Year. [Reference]	Weight Results	Adherence Results
He et al., 2017 [22]	Weight loss: the control group lost (−1.78 kg) and the intervention group (−2.09 kg).	-
	Significant weight loss at 6 months for men, but not for women (*p <* 0.001).	
	Men in WeChat group: higher probability of maintaining weight, Weight loss of 1 to 2 kg or Weight loss 1 of more than 2 kg than the control group.	
Spring et al., 2017 [23]	At 12 months, weight loss in standard group (−5.6 kg), in technology group (−3.1 kg), and in self (control) group (−2.7 kg).	Adherence to self-monitoring: larger in App group than in standard (*p <* 0.001) and it covaried with weight loss (r(84) = 0.36–0.51, *p <* 0.001). Correlations did not differ depending on the treatment condition.
	No significant differences between technology group and self-group (control).	
	(*p*-value not specified).	
Ross & Wing RR., 2016 [17]	Weight loss 6 months later: TECH + PHONE group (−6.4 kg), TECH group (−4.1 kg) and ST group (−1.3 kg).	Adherence to self-monitoring: greater in Tech + Phone group than in Tech and standard (*p <* 0.05), and standard group showed lower adherence. In both technology groups: Significant association between the percentage change in weight and adherence to intake control (r = −0.48, *p <* 0.001) and weight (r = −0.42, *p* = 0.002). No association between adherence to the use of the activity monitor and weight change (*p =* 0.085).
	Significant group differences (*p =* 0.042).	
Apiñaniz et al., 2019 [13]	No significant group differences in weight (0.357 kg, *p =* 0.7).	Increased adherence in App Group for dietary intake and PA recommendations. No significant group differences (dietary intake: *p =* 0.413 and PA: *p* = 0.145).
Wharton et al., 2014 [16]	Significant weight loss at 8 weeks in groups (app: −3.5 lb; Memo: −6.5 lb, and *diary*: −4.4 lb).	The App group registered more frequently than the paper group (*p =* 0.042).
	No significant group differences (*p* = 0.19).	
Thomas et al., 2019 [24]	Weight loss	The Smart Group (App) had the highest rate of weight self-control (3.7%) but did not differ significantly from the control group (29.7%).
	CONTROL: 18M −6.4 kg	
	GROUP: 18M −5.9 kg	
	SMART: 18M −5.5 kg	
	No significant group differences (*p* > 0.05)	
Svetkey et al., 2015 [25]	Weight loss: App group (−0.99 kg) vs. control group (−1.44 kg).	-
	No significant differences (*p*-value not specified)	
	Smartphone group (coaching) (−2.45 kg)	
	No significant differences between the App group and control. (*p*-value not specified).	
Allen et al., 2013 [26]	Weight loss 6 months later: intensive counseling (IC) (−2.5 kg); intensive counseling+smartphone (IC + SP) (−5.4 kg); less intensive counseling +smartphone (LIC + SP) (−3.3 kg); smartphone (SP) (−1.8 kg);	Larger in group intensive counseling + smartphone: they attended an average of 72% of the 14 counseling sessions and recorded their diet an average of 53% and PA 32% of the days. Very similar percentages for less intensive counseling group + smartphone and smartphone group.
	No significant group differences (*p* = 0.89).	
Turner-McGrievy & Tate., 2011 [27]	Weight loss 6 months later: podcast group (−2.6 kg) vs. podcast + mobile goup(−2.6 kg); Non-significant group differences (*p =* 0.88)	-
Lee et al., 2010 [28]	Changes in weight: Intervention group (−2.2 kg) vs. control group (−0.5 kg). (*p* < 0.05).	-
Laing et al., 2014 [29]	Weight, changes from the start:	-
	Control: 6M: + 0.27 kg	
	Intervention: 6M: −0.03 kg	
	Group differences: 6M: −0.30 kg (*p* = 0.63)	
Napolitano et al., 2013 [11]	Weight:	Neither the fast response time nor the number of text responses were associated with weight loss at week 4 or week 8.
	Facebook: 8 weeks: −0.63 kg	
	Facebook plus: 8 weeks: −2.4 kg	
	Control: 8 weeks: −0.24 kg	
	Significant group differences (8 weeks) (*p* < 0.05)	
Stephens et al., 2017 [30]	Weight, Smartphone + HC group: Beginning: 82.8 kg; 3 months: 8.1 kg. Difference: 2.1 kg.	-
	Control group: Beginning: 75.8 kg; 3 months: 77.3 kg. Difference: +1.5 kg.	
	Significant group differences (*p* = 0.026)	
Martin et al., 2015 [31]	Weight loss 12 weeks: the smartloss group lost (−7.08 kg) and the education group health care lost (−0.6 kg).	-
	Significant group differences (*p =* 0.001).	
Carter et al., 2013 [32]	Weight loss 6 months later: the App group lost (−4.6 kg), the *diary* group lost (−2.9 kg), and the web group lost (−1.3 kg).	Adherence was significantly higher in the App group (92 days) compared to the web group (35 days), and the paper group (29 days). (*p <* 0.001).
	No significant differences between App group and paper diary group (*p =* 0.12)	
Hernández-Reyes et al., 2020 [33]	Weight loss: intervention group lost (−7.9 kg), and the control group lost (−7.1 kg).	-
	No significant group differences (*p* > 0.05).	
Rogers et al., 2016 [34]	Weight loss 6 months later: the standard group lost (−6.57 kg), the improved tech group (en-tech) lost (−6.25 kg), and the tech group (−5.18 kg).	The standard group monitored their diet an average of 84.6 days, the technology group 80.0 days, and the enhanced technology group 70.1 days.
	No significant group differences (*p =* 0.0997).	
Alencar et al., 2019 [35]	Weight loss: the App group lost (−7.3 kg), and the control group lost (−1.5 kg).	-
	Significant group differences (*p <* 0.05).	
Lewis et al., 2019 [36]	Participants who started the intervention achieved significant decreases in: weight (−4.87 kg) at 4 months, maintaining these losses after switching to the control group.	The addition of telephone and texting support to a community obesity management program improved behavioral adherence compared to standard care.
	In participants who started the control group, no significant changes were observed at 4 months. After the intervention, significant reductions were achieved in: weight (−2.76 kg), at 8 months.	
	Significant group differences (*p =* 0.01)	
Steinberg et al., 2013 [37]	Changes in weight: Control group participants gained an average of 1.14 kg, while intervention group lost an average of 1.27 kg.	Trend towards greater adherence to text messages associated with a higher percentage of weight loss (r = −0.36, *p =* 0.08), but this did not reach statistical significance.
	Significant group differences (*p =* 0.09)	
Shapiro et al., 2012 [38]	Weight loss 12 months later: control group (−2.27 lb) and intervention group (−3.64 lb); control group lost an average of 0.8% of the weight, and the intervention group 1.8%.	Adherence to text messages was moderately strong (60-69%). Participants with higher adherence lost more weight at 6 (*p =* 0.039) and 12 months (*p =* 0.023) than those who adhered less.
	Non-significant group differences (*p =* 0.394).	
Jakicic et al., 2016 [39]	Weight loss 24 months later: the standard group (SBWI) lost −5.9 kg and the technology group (EWLI) lost −3.2 kg.	-
	Significant group differences (*p =* 0.002).	
Godino et al., 2019 [40]	Weight loss 12 months later: control group (−.61%), SMS group (−1.6%), and SMS+Coaching group (−3.63%).	The median of the average daily commitment rate decreased slightly over time: 28.69 at 6 months and 24.91 at 12 months. A unit increase in the average percentage of daily participation throughout the study was associated with a higher percentage of weight loss (−0.08%, *p <* 0.05).
	At 12 months, Weight loss the average percentage, adjusted for baseline BMI, was significantly different between SMS + coaching and the control group −3.0, but not between SMS alone and the control group −1.07; (*p =* 0.291).	
Newton et al., 2018 [41]	Weight loss: intervention group lost (−1.4 kg), and the control group gained (0.2 kg).	The correlation between the number of SMS text messages sent and the change in weight loss was not statistically significant.
	Significant group difference in the Weight loss (*p =* 0.03).	
Bouhaidar et al., 2013 [42]	Weight loss: intervention group lost (−5.96 lb) and the control group (−1.41 lb).	At the beginning of the intervention, participants’ response rate to SMS requests was 66%. This percentage decreased to 52% at the end of the intervention.
	Significant group differences (no *p*-value is specified).	
Lin et al., 2014 [43]	Weight changes at 6 months: g. intervention lost (−1.6 kg) and control group gained (+0.24 kg), with a group difference of 1.83 kg. Significant group differences (*p <* 0.0001)	No significant correlation between weight loss at 6 months and total follow-up days (*r* = 0.14, *p =* 0.27), nor did it correlate significantly with the average percentage of follow-up days (*r* = 0.14, *p =* 0.27).
Haapala et al., 2009 [44]	Weight loss at 12 months, the intervention group lost (−4.5 kg), and the control group (−1.1 kg)	The overall frequency of use of the program decreased from 8 times per week to 3–4 times per week in 12 months. Those with more than 5% weight loss at 12 months reported more frequent weekly contact at 3 months than those who had lost less than 5%.
	Significant group differences (*p =* 0.006).	
Patrick et al., 2009 [45]	Intervention group: Beginning: 89.79 kg; Month 4: 85.17. Difference: 4.62 kg.	During the first week, participants responded to all messages that requested a response. At week 16, participants responded to approximately two out of every three messages.
	Comparison group: Beginning: 88.02 kg; Month 4: 87.85 kg. Difference: 0.17 kg).	
	Weight loss adjusted comparison group vs. Weight loss adjusted intervention group (−2.88 kg).	
	Significant group differences (*p =* 0.02)	
Sullivan et al., 2013 [46]	Weight loss: the face-to-face group lost (−1.8%), and the virtual reality group (−7.6%).	-
	Significant group differences (*p <* 0.05).	
	Weight maintenance: virtual reality group (14%) compared to face-to-face group (9.5%). Significant group differences (*p <* 0.05).	
Behm-Morawitz et al., 2016 [47]	Weight Loss: experimental groups lost (−1.75 lb), and the control group (−0.91 lb). Significant group differences (*p =* 0.04).	-
Johnston et al., 2012 [15]	Weight loss: virtual world group (−3.9 kg) and face-to-face group (−2.8 kg).	-
	No significant group differences (*p =* 0.29).	
Manzoni et al., 2016 [12]	Weight loss: the SBP group lost (−6.2%), the CBT group lost (−7.4%) and the virtual reality group lost (−6.25%).	-
	No significant group differences (*p* > 0.05)	
	The RV group is more likely to maintain or further improve weight loss at the 1-year follow-up than the SBP group (48% vs. 11%; *p =* 0.004) and than the CBT group (48% vs. 29%; *p =* 08).	
Morgan et al., 2014 [48]	Weight loss 6 months later: the online group lost (−4.7 kg) and the resources group (pedometer) (−3.7 kg) compared to the control group (−0.5 kg).	The number of objectives established (β = −0.3, 95% CI [−0.6, −0.1], *p =* 0.01) and the number of documented weight records (β = −0.21, 95% CI [−0.39, −0.02], *p =* 0.03) were independently associated with weight loss, so each additional goal set and the recorded weight measurement were associated with increased weight loss of 0.32 kg and 0.21 kg, respectively (R^2^ = 0.37, *p <* 0.001).
	Significant differences between the two intervention groups with technology and the control group (*p <* 0.001) but no significant differences in the intervention groups (*p* > 0.05).	
Azar et al., 2015 [49]	Weight loss: the intervention group lost (−3.6 kg), and the control group (−0.4 kg). Intervention group lost on average 3.2 kg more than the control group.	While not statistically significant, the downward slope of both the assistance/weight loss and self-monitoring/weight loss curves suggests a weight loss trend with greater participation.
	Significant group differences (*p =* 0.0002).	
Abdi et al., 2015 [50]	Weight:	−
	Web Group: Beginning: 79.44 kg; Month 9: 78.04 kg. Difference: 1.4 kg.	
	Phone Group: Beginning: 83.01 kg; Month 9: 82.02 kg. Difference: 0.99 kg.	
	Control group: Beginning: 78.63 kg; Month 9: 78.83 kg. Difference: +0.2 kg.	
	Significant group differences (*p =* 0.001).	
Morgan et al., 2011 [51]	Weight loss 12 months later: web group lost (−5.3 kg), and the control group lost (−3.1 kg)	Significant correlation between weight change at 12 months and the number of days of diet entries (weight: *r* = 0.69, *p <* 0.001), number of diary entries (weight: *r* = 0.54, *p =* 0.004) and number of weekly weight entries (weight: *r* = 0.56, *p =* 0.004).
	No significant group differences (*p =* 0.408).	
Burke et al., 2012 [52]	Weight loss 24 months later: the PDA+FEEDBACK group lost (−2.17 kg), the paper group lost (−1.77 kg), and the PDA group lost (−1.18 kg).	Significant differences between PDA and PDA + Feedback groups and paper group (*p =* 0.03). No significant differences between PDA + Feedback groups and PDA group (*p =* 0.49). A higher proportion of the PDA groups, compared to the paper group was adherent 60% or more of the time (PDA + Feedback vs. paper, *p =* 0.01) and (PDA vs. paper, *p* = 0.03). 18 months: 19−20% of PDA groups remained adherent 30% or more of the time, compared to 8% of the paper group.
	No significant group differences (*p =* 0.33).	
Wang et al., 2012 [53]	Weight loss 12 months later: paper diary group (−5.19 lb), PDA group (−3.92 lb) and PDA+FB group (−5.30 lb)	Compared to paper register, the PDA to control diet (*p =* 0.027) and PA (*p =* 0.014) had significant direct effects on weight loss at 12 months. And a significant indirect effect on results through better adherence to self-monitoring (*p <* 0.001).
	No reference to whether there are significant group differences.	
Chung et al., 2014 [54]	Weight loss:	-
	Control: Beginning: 71.7 kg; SEM.12: 7 kg. Difference (−1.5 kg)	
	“Food diary”: Beginning: 71.4 kg; SEM.12: 69.7 kg. Difference (−1.7 kg)	
	“Electronic diary”: Beginning: 7 kg; SEM.12: 68.4 kg. Difference (−1.6 kg)	
	No significant differences between the intervention groups and the control group (*p =* 0.159).	
Spring et al., 2013 [55]	Weight loss12 months later: the PDA group lost (−2.9 kg), and the standard group lost (−0.02 kg).	No difference in how often participants in any group attended the MOVE! groups. (Group +mobile: mean of 6.2 meetings vs. Standard Group: mean of 5.9 meetings (*p =* 0.54). The +mobile participants who attended 80% or more of the treatment sessions lost significantly more weight than the less adherent participants of the +mobile group and, than the adherent or non-adherent participants of the standard group.
	Significant group differences (no *p*-values).	
Unick et al., 2012 [56]	Weight loss: standard group + technology (SBWL + tech) lost (−10%), and the standard group (SBWL) lost (−7.8%).	The self-monitoring of food intake was considerably higher in technology compared to standard group (86.2% vs. 71.5%), but did not reach significance (*p =* 0.098). The technology group wore the bracelet for 91.3% of the days.
	No significant group differences (*p =* 0.46).	
Shuger et al., 2011 [57]	Weight loss:	Adherence to wearing the bracelet was greater than 55%, suggesting that weight loss participants may adhere better to self-monitoring protocols that use technology, compared to standard protocols.
	Standard group: beginning: 102.2 kg; Month 9: 101.32 kg. Difference: 0.88 kg.	
	GWL group: beginning: 101.84 kg; Month 9: 99.98 kg. Difference: 1.86 kg.	
	SWA group: beginning: 101.15 kg; Month 9: 97.60 kg. Difference: −3.55 kg	
	GWL + SWA group: beginning: 10.32 kg; Month 9: 93.73 kg. Difference: −6.59 kg	
	Significant group differences (*p =* 0.0001). Difference:	
Pellegrini et al., 2012 [58]	Weight loss: SBWL group: beginning: 88.6 kg; Month 6: 84.9 kg. Difference: 3.7 kg.SBWL + TECH group: beginning: 102.1 kg; Month 6: 93.3 kg. Difference: 8.8 kg.	SBWL+TECH group self-monitors dietary intake significantly more days (5.9 days/week) than SBWL group (5.3 days/week; *p <* 0.05) or the TECH group (5.2 days/week; *p =* 0.001). The self-monitoring of dietary intake was significantly related to weight loss at 6 months when the intervention groups were combined (*r* = −0.57, *p <* 0.001) and when analyzed separately for TECH (r = −0.64, *p =* 0.02).
	TECH group: beginning: 92.3 kg; Month 6: 86.4 kg. Difference: 5.9 kg	
	Significant group differences (*p =* 0.044)	
Dunn et al., 2014 [59]	Weight loss:	-
	Online group: Beginning: 197.85 lb; End: 189.81 lb. Difference: 8.04 lb.	
	In-person group: Beginning: 197.02 lb; End: 191.07 lb. Difference: 5.95 lb	
	Significant differences (*p =* 0.02)	
Chambliss et al., 2011 [60]	Weight loss: Basic and Improved groups lost (−2.7 kg and −2.5 kg, respectively) and the Control group (−0.3 kg)	-
	Significant differences (*p <* 0.05).	
Ing et al., 2018 [61]	Weight Loss: the DVD group lost (−0.48 kg) and face-to-face group (−0.07 kg).	-
	Non-significant differences (*p =* 0.46).	
	Weight Maintenance: 64.5% in the DVD group and 52.4% in the face-to-face group.	
	No significant differences (*p =* 0.58).	
Godino et al., 2016 [62]	No significant differences in weight between groups (−0.79 kg, *p =* 0.204).	-

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
