# Peer review of "Use of Technology-Based Interventions in the Treatment of Patients with Overweight and Obesity: A Systematic Review"

_nutrients, 2020, doi:10.3390/nu12123634_

Round 1

Reviewer 1 Report

This paper resents the results of a systematic review on the impact of technology-based interventions on the efficacy/effectiveness of weight-loss interventions in overweight and obese adults. The authors also intend to assess in how far such interventions improve adherence to dietary nutritional treatment keeping in mind the long term effectiveness. With that, the review addresses an important issue in overweight and obesity interventions, being the problem of sustainable behaviour change.

Broad comments

This review touches on an the interesting subject of the (additional) value of technology based approaches in overweight and obesity interventions. The review seems to be executed well and provides interesting results. However, I miss the announced quality (and risk of bias) assessment of the included articles. Also a sensitivity analysis may add to the value of the findings, because it appears that randomised and quasi experimental designs have been included.

My main question is about the evidence for long term effectiveness. In their introduction the authors rightly mention the problem of the declining effect over time of overweight and obesity interventions. They seem to make their case, at least to a certain extent, about the value of technology based interventions for long term effectiveness. Unfortunately, the results section provides very limited information about that subject. Only presenting information about better adherence (how was that measured?) during the intervention period doesn’t provide evidence for a better long term effect.

Specific comments

METHODOLOGY

2.4 Inclusion and exclusion criteria (p3)

articles that presented at least two groups for comparison,..

Does this include only RCTs or also quasi experimental designs?

The review intends to assess in how far such technological interventions improve adherence to dietary nutritional treatment. However, I see no measure of adherence. Could the authors explain why a measure for adherence was not an inclusion criterion?

2.5 Extracted data (p 3)

Was a predefined data extraction form used and if so, which one?

How was adherence to dietary nutritional treatment measured?

RESULTS

3.1 Descriptive data and types of studies (p 4)

Various designs are described here. Taking the inclusion criteria into account they should all include a control or comparison group. From how I read it now, I identify 42 RCTs (26+10+6) of which 6 with low power (pilots, probably with small n). Or are the 10 randomised trials not RCTs (what is for instance a randomised pretest-posttest design?) ? Unless the cohort study includes a nested RCT and the mixed methods study includes a RCT there seem to be 7 quasi experiments. Is that correct? This would bring the total to 49 included studies.

It looks like ref 22 and 26 are two different publications describing the same study.

3.3.1 Smartphone (p 17)

Among the studios that used a smartphone, 18 of them used an App, and the other 10 used

text messages and/or calls.

This information doesn’t seem to correspond with the numbers presented in paragraph 3.2.1. Smartphone. If the information is correct, I think it should be presented in that paragraph.

  • APP (p 17)

Of eleven studies reporting weight loss, 6 showed significant differences in favour of the intervention group, so these six are the only studies providing evidence for effectiveness off the app. The way it is written now suggests evidence for effectiveness in more studies.

In seven articles, adherence was greater in the groups that used or complemented the

intervention with an App,

A description of how adherence was measured is lacking. Were valid measurement instruments used? Otherwise it remains unclear what we can learn from this review as regards evidence for the impact of technology-based interventions on adherence.

  • SMS and or Calls (p 17)

Here also only 7 of 9 studies provide evidence for an effect of the technology based approach.

3.3.4. Personal digital assistant (pda) or electronic journal (ej) (p 18)

It remains a bit unclear how many of the pda and ej studies provided evidence for effectiveness, meaning significantly larger improvements than the control group. Could the authors clarify that?

DISCUSSION

Lines 2-4

The results found in this work indicate that weight loss and/or maintenance was greater in the

groups whose intervention was performed or complemented by one of the aforementioned types of technology (p 32)

Reading the Result section I cannot agree with the conclusion that maintenance was greater. When I am correct only one article included a long term follow up with actual measurement of weight loss. I haven’t found any sound evidence for this assumption in the rest of the results section (see also my broader comment to the study). Please reconsider.

As regards the effectiveness it seems that the evidence for technology based interventions differs between the types of such interventions. Maybe the authors could make a distinction between types of interventions.

Lines 19-21

For example, Raaijmakers et al. [63] found that half of the technology-based nterventions (54%) significantly helped participants lose weight, compared to the lack of attention or habitual attention.

What is meant by lack of attention or habitual attention?

Lines 28-30

The provision of feedback also appeared to be effective as a complement to interventions carried

out using technology to promote weight loss, as can be seen in three of the studies analyzed [22, 26 30].

The authors don’t seem to emphasize the added value of feedback very strongly in their Results section. So, unless the evidence for feedback is stronger than presented in the Result section, this statement about the effectiveness of feedback may need to be phrased more cautiously.

Lines 43/44

Of all the methods analyzed, physical activity monitors were the type of technology that achieved the greatest weight loss

Based on what evidence do the authors draw that conclusion, because they do not present specific data on the results of PA monitors in their Results section? Please provide the evidence for this statement?

Lines 48-55

The authors rightly mention 6 limitations but forget to indicate what the consequences of these limitations are. Could they add that?

CONCLUSION

In my opinion, the authors should reconsider whether the findings of this review warrant a conclusion in such absolute terms.

TABLES

Please revise Table 3, something caused a shift starting at Carter et al. 2013

Typos

Summary

first sentence ‘…health problems worldwide’

Results

3.2.4. Personal digital assistant (pda) or elelctric journal

3.3.1. Smartphone

Among the studios that used a smartphone

Author Response

Comments and Suggestions for Authors

This paper resents the results of a systematic review on the impact of technology-based interventions on the efficacy/effectiveness of weight-loss interventions in overweight and obese adults. The authors also intend to assess in how far such interventions improve adherence to dietary nutritional treatment keeping in mind the long term effectiveness. With that, the review addresses an important issue in overweight and obesity interventions, being the problem of sustainable behaviour change.

Broad comments

This review touches on an the interesting subject of the (additional) value of technology based approaches in overweight and obesity interventions. The review seems to be executed well and provides interesting results. However, I miss the announced quality (and risk of bias) assessment of the included articles. Also a sensitivity analysis may add to the value of the findings, because it appears that randomised and quasi experimental designs have been included.

My main question is about the evidence for long term effectiveness. In their introduction the authors rightly mention the problem of the declining effect over time of overweight and obesity interventions. They seem to make their case, at least to a certain extent, about the value of technology based interventions for long term effectiveness. Unfortunately, the results section provides very limited information about that subject. Only presenting information about better adherence (how was that measured?) during the intervention period doesn’t provide evidence for a better long term effect.

First of all, we would like to thank you for your comments and suggestions because they can help us to improve the quality of our systematic review.

Specific comments

METHODOLOGY

2.4 Inclusion and exclusion criteria (p3)

…articles that presented at least two groups for comparison,..

Does this include only RCTs or also quasi experimental designs?

This includes both RCTs and quasi experimental designs.

The review intends to assess in how far such technological interventions improve adherence to dietary nutritional treatment. However, I see no measure of adherence. Could the authors explain why a measure for adherence was not an inclusion criterion?

Thank you very much for your suggestion. Now, we have modified the aim of the study according to your comment. The adherence to nutritional treatment was not included as an inclusion criterion due to most studies did not examine this question. We were interested in assessing how different types of technologies, mobile phones, PDA internet based tools, social networks, smartphones and their applications, and virtual reality may play an important role on weight loss in overweight and obese patients (Lines 78-80).

2.5 Extracted data (p 3)

Was a predefined data extraction form used and if so, which one?

The quality of each primary study was assessed with the Cochrane Collaboration Risk of Bias (ROB) tool (Higgins et al., 2011), which includes seven items covering six domains of bias. Each item is judged as having a high, low, or unclear ROB. We have included this information into manuscripts in lines (81-96).

How was adherence to dietary nutritional treatment measured?

Studies do not use a questionnaire as such to measure adherence, because there is no instrument to measure adherence to nutritional dietary treatment; what they do here is measure adherence to self-monitoring. According to a recent study, the potential of technologies to facilitate weight loss lies in their ability to increase adherence to treatment through strategies such as self-monitoring (Dounavi & Tsoumani., 2019; PMID: 31003801). And, adherence to self-monitoring has been defined as the record of at least the amount of food that would be equivalent to 50% of the energy goal for the day (Burke et al., 2008; PMID: 21185970).

RESULTS

3.1 Descriptive data and types of studies (p 4)

Various designs are described here. Taking the inclusion criteria into account they should all include a control or comparison group. From how I read it now, I identify 42 RCTs (26+10+6) of which 6 with low power (pilots, probably with small n). Or are the 10 randomised trials not RCTs (what is for instance a randomised pretest-posttest design?) ? Unless the cohort study includes a nested RCT and the mixed methods study includes a RCT there seem to be 7 quasi experiments. Is that correct? This would bring the total to 49 included studies.

Thank you for your comment, we have revised every article to see what the design of each one was and we have changed the way it was written because we made some mistakes. For example, there was no mixed methods study, there were two studies from the same author (Napolitano et al., 2013) and we counted the mixed methods one which was excluded, instead of the randomized controlled trial from this same author. Another thing is that there were a total of 49 studies, which was wrong, because we counted twice the same article (Laing et al., 37). We have written this paragraph as follows (Lines 155-159): Table 2, which also lists the design of the studies, shows that of the 47 studies included, 30 of them were randomized controlled trials [11, 12, 13, 21, 23-25, 27, 29, 32, 36, 38, 39-41, 43-48, 50-56, 60, 61]. Six were randomized pilot studies [17, 28, 30, 33, 37, 55] and one was a non-randomized pilot study (16). Six were randomized pretest-posttest design [22, 26, 34, 42, 49, 57] and four were quasi experimental designs [15, 31, 58, 59].

It looks like ref 22 and 26 are two different publications describing the same study.

Thank you for this comment, we have revised it and you are absolutely right. References 22 and 26 are the same study, so we will solve this problem excluding reference 22 because of duplication. Consequently, after removing reference 22, we have corrected the numbers of each of the references. For example, the reference that was the number 23, is now the number 22 and so on with all the following.

3.3.1 Smartphone (p 17)

Among the studios that used a smartphone, 18 of them used an App, and the other 10 used text messages and/or calls.

This information doesn’t seem to correspond with the numbers presented in paragraph 3.2.1. Smartphone. If the information is correct, I think it should be presented in that paragraph.

Thank you for your comment, we have revised this information and it is correct. As you suggested, to make it clearer, we have included this information in paragraph 3.2.1 as follows as (Line 165-171): In eighteen studies a mobile App (App) was used in their intervention; eleven of them used an App to monitor intake, weight, and/or physical activity (11, 13, 24, 30 36, 39, 44, 49, 58, 59). Five articles, in addition to an App to monitor intake and weight, used a physical activity monitor (17, 38, 42, 43, 53). One article used the mobile app and/or the website (55), and in another article, two apps were used, one for self-monitoring and one for video-conferences (47). In the other 10 articles, the intervention was carried out by telephone calls and/or SMS (21, 27, 31, 33, 46, 48, 52, 54, 57, 61).

APP (p 17)

Of eleven studies reporting weight loss, 6 showed significant differences in favour of the intervention group, so these six are the only studies providing evidence for effectiveness off the app. The way it is written now suggests evidence for effectiveness in more studies.

We thank you for your comment, we have improved the way it is written the information so that there aren`t misunderstandings while reading the article. Lines (194-196): Six of the articles that used or supplemented the intervention with an App exclusively or together with some physical activity monitor reported evidence of weight loss compared to the control or comparison groups (Table 4)…

In seven articles, adherence was greater in the groups that used or complemented the intervention with an App,

A description of how adherence was measured is lacking. Were valid measurement instruments used? Otherwise it remains unclear what we can learn from this review as regards evidence for the impact of technology-based interventions on adherence.

Studies do not use a questionnaire as such to measure adherence, because there is no instrument to measure adherence to nutritional dietary treatment; what they do here is measure adherence to self-monitoring. According to a recent study, the potential of technologies to facilitate weight loss lies in their ability to increase adherence to treatment through strategies such as self-monitoring (Dounavi & Tsoumani., 2019; PMID: 31003801). And, in a recent report from a clinical trial, adherence to self-monitoring has been defined as the record of at least the amount of food that would be equivalent to 50% of the energy goal for the day (Burke et al., 2011; PMID: 21185970).

SMS and or Calls (p 17)

Here also only 7 of 9 studies provide evidence for an effect of the technology based approach.

We thank you for your comment, we have improved the way it is written the information so that there aren`t misunderstandings while reading the article. Lines (211-212): Seven of the studies analyzed that used SMS or calls found a significant effect of their use, with significant weight loss compared to the control or comparison groups…

3.3.4. Personal digital assistant (pda) or electronic journal (ej) (p 18)

It remains a bit unclear how many of the pda and ej studies provided evidence for effectiveness, meaning significantly larger improvements than the control group. Could the authors clarify that?

We thank you for this comment, you are right it is not very clear how many of the pda and ej studies provided evidence for effectiveness so we will try to clarify that. Among the four studies that used this type of technology, just in one study the differences between groups were significant. Because of that, we have modified the way it is written to make it clearer (Lines 243-244): Among the four studies that used this type of technology, just in one of them, the PDA group lost significantly more weight than the control group (-2.9 kg vs. - .02 kg; p = n.s.) [32].

DISCUSSION

Lines 2-4

The results found in this work indicate that weight loss and/or maintenance was greater in the groups whose intervention was performed or complemented by one of the aforementioned types of technology (p 32)

Reading the Result section I cannot agree with the conclusion that maintenance was greater. When I am correct only one article included a long term follow up with actual measurement of weight loss. I haven’t found any sound evidence for this assumption in the rest of the results section (see also my broader comment to the study). Please reconsider.

We totally agree with the reviewer´s suggestion. Now, we have changed it into discussion section as follows as, (Line 259-263): The results found in this work indicate that weight loss was greater in the groups whose intervention was performed or complemented by one of the aforementioned types of technology, although in 13 studies, the differences with the control or comparison groups were not statistically significant [13, 15, 25, 27, 28, 30, 36, 41, 50, 53, 55, 56].  However, the same cannot be concluded regarding weight maintenance, since most of the studies did not include this outcome.

We have also modified a sentence where we suggested a similar idea in the following way (Lines 272-274): In general, these results suggest that the use of different types of technology for self-monitoring of diet, physical activity, and/or weight is effective in promoting weight loss among people who are overweight or obese.

As regards the effectiveness it seems that the evidence for technology based interventions differs between the types of such interventions. Maybe the authors could make a distinction between types of interventions.

Thank you, for your suggestion. We have taken it into account, the information is in lines (303-305) where we classify the types of technology from highest to lowest average weight loss: Of all the methods analyzed, physical activity monitors were the type of technology that achieved the greatest weight loss (M=6.21kg), followed by virtual reality (4.7kg), website (3.75kg), smartphone (3.44kg), and PDA (2.0kg).

Lines 19-21:

For example, Raaijmakers et al. [63] found that half of the technology-based interventions (54%) significantly helped participants lose weight, compared to the lack of attention or habitual attention.

What is meant by lack of attention or habitual attention?

We have looked up what was meant by lack of attention or habitual attention; lack of attention was when a control group did not receive any kind of intervention and habitual attention consisted mostly of a lifestyle intervention and/or counselling without technology aspects.

Lines 28-30

The provision of feedback also appeared to be effective as a complement to interventions carried out using technology to promote weight loss, as can be seen in three of the studies analysed [22, 26 30].

The authors don’t seem to emphasize the added value of feedback very strongly in their Results section. So, unless the evidence for feedback is stronger than presented in the Result section, this statement about the effectiveness of feedback may need to be phrased more cautiously.

First of all, thank you for this comment. We think you are right and this statement should be phrased more cautiously. Lines (287-291): The provision of feedback could be effective as a complement to interventions carried out using technology to promote weight loss [25, 29]. This might suggest that receiving feedback in the form of text messages or emails could improve adherence to self-monitoring and, as a result, lead to increased weight loss. Nevertheless, more research is needed on this topic since the evidence found in this systematic review is not strong enough.

Lines 43/44

Of all the methods analyzed, physical activity monitors were the type of technology that achieved the greatest weight loss

Based on what evidence do the authors draw that conclusion, because they do not present specific data on the results of PA monitors in their Results section? Please provide the evidence for this statement?

We have included information in the results section (3.3.5 other types of technology; Lines 253-255: Physical activity monitors were the type of technology that achieved the highest weight loss (6.21kg)…). And if you compare that information with the average weight loss achieved by Smartphone (3.44 kg), virtual reality (4.7 kg), website (3.75 kg) and PDA or EJ (2.0kg), the physical activity monitors were the type of technology that achieved the highest average weight loss (6.21kg), so our results suggest that from other types of technology, physical activity monitors could be a useful strategy for weight loss. However, as we say in lines (305-307): only 7 studies of those analyzed used these types of technology to perform the intervention, and therefore, it is impossible to know exactly whether these mean weight losses would remain so high after being evaluated in more groups of people.

Lines 48-55

The authors rightly mention 6 limitations but forget to indicate what the consequences of these limitations are. Could they add that?

Thank you for your comment; we have added the consequences of each one of the limitations (Lines 308-315)

First, the wide variability in the design of the studies included limits the conclusions that can be reached. Second, the search only included English and Spanish publications, which may not have represented all the available evidence. Thirdly, heterogeneity of the time periods of the intervention was high, ranging from a few weeks to 24 months, which can affect the strength of our results and conclusions. Fourth, the presence of studies that used a small sample size may be associated with greater uncertainty about the measured effect. And, fifth, heterogeneity in the type of intervention performed and the groups with which the comparison was made, which can make the comparison of effectiveness difficult to investigate.

CONCLUSION

In my opinion, the authors should reconsider whether the findings of this review warrant a conclusion in such absolute terms.

Thank you for your comment, we have modified the conclusion, trying not to make conclusions in such absolute terms. Lines (320-328)Weight loss programs for people who are overweight or obese, carried out or supplemented by some kind of technology, seem to lead to greater weight loss compared to traditional programs. Physical activity monitors and virtual reality were the types of technology that lead to increased weight loss, although further research is needed on the use of these types of technology, as the evidence found is scarce. The use of technology also seems to allow improvement in adherence to treatment, as it allows a simpler and faster self-monitoring. In addition, although more research is needed, this could improve more when the technology is accompanied by immediate feedback. However, future research should focus on this, as, despite the use of technology, adherence to dietary-nutritional treatment often decreases over time, resulting in less weight loss as time passes.

TABLES

Please revise Table 3, something caused a shift starting at Carter et al. 2013

Thank you, the shift has been corrected.

Typos

Summary

First sentence ‘…health problem worldwide’  health problems worldwide

Results

3.2.4. Personal digital assistant (pda) or elelctric journal  electronic journal

3.3.1. Smartphone

Among the studios that used a smartphone the sentence where this typo was has been deleted.

Reviewer 2 Report

This review discusses an important issue of obesity treatment. Use of technology has recently increased and seen to have beneficial effects. A recent study (pubmed id 31080628) has investigated intervention for weight loss based on technology and biomarker information. This has allowed to provide biochemical based feedback with the use of technology leading to improved lifestyle choices. Another study by S. Berry et.al has also shown that personalised nutrition can be beneficial for interventions for losing weight. How do you think the use of technology with biological information would be beneficial for treatment of patients with obesity?

Author Response

Reviewer 2

Comments and Suggestions for Authors

This review discusses an important issue of obesity treatment. Use of technology has recently increased and seen to have beneficial effects. A recent study (pubmed id 31080628) has investigated intervention for weight loss based on technology and biomarker information. This has allowed to provide biochemical based feedback with the use of technology leading to improved lifestyle choices. Another study by S. Berry et.al has also shown that personalised nutrition can be beneficial for interventions for losing weight. How do you think the use of technology with biological information would be beneficial for treatment of patients with obesity?

First of all, we thank you for your comment and suggestion. We consider that biomarker information is very important, because it can give us important information about a person, from which their diet could be personalize, and consequently lead to weight loss in obese patients. Although, we have not taken them into account for this research but given their importance we will take them into account for future ones, since as some studies have said, they seem to can provide a quantitative guide to personalised diet and physical interventions.

Round 2

Reviewer 2 Report

Accept in the present form